# Generative Inbetweening: Adapting Image-to-Video Diffusion models for Keyframe Interpolation

**Xiaojuan Wang**[1]   **Boyang Zhou**[1]   **Brian Curless**[1]   **Ira Kemelmacher-Shlizerman**[1]
**Aleksander Holynski**[2,3]   **Steven M. Seitz**[1]
[1]University of Washington, [2]Google DeepMind, [3]UC Berkeley

## Abstract

We present a method for generating video sequences with coherent motion between a pair of input keyframes. We adapt a pretrained large-scale image-to-video diffusion model (originally trained to generate videos moving forward in time from a single input image) for keyframe interpolation, i.e., to produce a video between two input frames. We accomplish this adaptation through a lightweight fine-tuning technique that produces a version of the model that instead predicts videos moving *backwards* in time from a single input image. This model (along with the original forward-moving model) is subsequently used in a dual-directional diffusion sampling process that combines the overlapping model estimates starting from each of the two keyframes. Our experiments shows that our method outperforms both existing diffusion-based methods and traditional frame interpolation techniques.

## 1 Introduction

Recent advances of large-scale text-to-video and image-to-video models (Blattmann et al., 2023b;a; Wu et al., 2023; Xing et al., 2023; Bar-Tal et al., 2024; Zeng et al., 2024) have shown the ability to generate high resolution videos with dynamic motion. While these models can accept a variety of input conditioning signals, such as text captions or single images, most available models remain unsuitable for an obvious application: keyframe interpolation. Interpolating between a pair of keyframes—that is, producing a video that simulates coherent motion between two input frames, one defining the starting frame of the video, and one defining the ending frame—is certainly possible if a large-scale model has been trained to accept these particular two conditioning signals, but most open-source models have not. Despite the task's similarity to existing conditioning signals, creating an interpolation model requires further training, and therefore both large amounts of data and substantial computational resources beyond what most researchers have access to.

Given the similarity between the input signals needed for keyframe interpolation (i.e., two-frame conditioning) and the input signals to existing models (e.g., one-frame conditioning), an interesting alternative solution is to instead *adapt* an existing pre-trained image-to-video model, without training a specialized model from scratch. In this paper, we propose an approach for enabling keyframe interpolation by doing precisely this. Our approach is founded upon the observation that a keyframe interpolation model needs to know how to accomplish three objectives: (1) given a starting frame, it needs to predict coherent motion starting from that frame and advancing into the *future*, (2) given an ending frame, it needs to predict coherent motion starting from that frame and advancing backwards into the *past*, and (3) given these two predictions, produce a video that has a coherent combination of the two. Since existing image-to-video models can already accomplish the first of these three objectives, we focus our efforts on the the latter two, i.e., producing a single-frame conditioned model that can generate motion *backwards* in time, and a mechanism for combining forward and backward motion predictions into coherent videos.

One may imagine that producing such a single-image conditioned model that produces backwards motion should be trivial: simply pass an image into a regular image-to-video model, and reverse the output. Unfortunately, real-world motion is inherently asymmetric, and reversed motion into the

future is notably different from motion into the past. As such, we first propose a novel, lightweight fine-tuning mechanism that reverses the arrow of time by rotating the temporal self-attention maps (i.e., reversing the temporal interactions) within the diffusion U-Net. This enables the reuse of the existing learned motion statistics in the pretrained model, and enables generalization while only requiring a small number of training videos.

Given both the original image-to-video model and this adapted reverse model, we also propose a sampling mechanism that merges the scores of both to produce a single consistent sample. These two sampling paths are synchronized through shared rotated temporal self-attention maps, ensuring they generate exactly opposite motions, an effect which we term "forward-backward motion consistency". At each sampling step, their intermediate noise predictions are fused, resulting in a generated video with coherent motion that starts and ends with the provided frames. We compare our work qualitatively and quantitatively to related methods on two curated difficult datasets targeted for generative inbetweening: Davis (Pont-Tuset et al., 2017) and Pexels[1], and our method produces notably higher quality videos with more coherent dynamics given distant keyframes.

## 2 RELATED WORKS

**Frame interpolation** Frame interpolation (Dong et al., 2023) synthesizes intermediate images between two frames by taking a pair of input frames or multiple adjacent frames in the context of video frame interpolation, and has been a long-standing research area in computer vision. Example applications include temporal up-sampling to increase refresh rate, create slow-motion videos, or interpolating between near-duplicate photos. Much of the research in this field (Jiang et al., 2018; Niklaus & Liu, 2020; Huang et al., 2020; Park et al., 2020; Lee et al., 2020; Park et al., 2021) employs flow-based methods, which estimate optical flow between the frames and then synthesize the middle images guided by the flow via either warping or splatting techniques. There are also works (Kalluri et al., 2023; Shi et al., 2022) that use CNNs or transformers to learn to extract features and directly output the middle frames. Traditionally, this task assumes unambiguous motion and the input frames are usually closely spaced ($\leq 1/30$s) samples in the video. Recent studies have begun to address large motions (Sim et al., 2021; Reda et al., 2022), or quadratic motion (Xu et al., 2019; Liu et al., 2020), though these still involve a single motion interpolation and cannot address distant input frames. In contrast, we aim to generate in-between frames that capture dynamic motions across distant input keyframes ($\geq 1$s apart) with a generative model, a challenge that goes beyond the capability of current frame interpolation techniques.

**Diffusion models for in-between video generation** Diffusion models have shown remarkable capabilities for generative modeling of images (Ho et al., 2020; Dhariwal & Nichol, 2021; Song & Ermon, 2019; Song et al., 2020a;b; Sohl-Dickstein et al., 2015) and videos (Ho et al., 2022b;a; Wu et al., 2023; Blattmann et al., 2023b). Early work MCVD (Voleti et al., 2022) devises a general-purpose diffusion model for a range of video generative modeling tasks including in-between video generation. More recent works (Guo et al., 2023; Jain et al., 2024; Xing et al., 2023) explicitly train diffusion models to accept two end frames with conditioning to generate 7 or 16 intermediate frames at maximum resolution of $320 \times 512$ at once, and achieved superior results in generating dynamic motions. In this work, we focus on *adapting* a pre-trained large-scale image-to-video model to do keyframe inbetweening without having to train or fine-tune from scratch. Exposed to millions of videos, these models have demonstrated remarkable capabilities in generating high-resolution (up to 1080p) and long (up to $4s$) videos with rich motion priors.

**Diffusion sampling for consistent generation** In diffusion-based image generation tasks, novel joint diffusion sampling techniques (Bar-Tal et al., 2023; Zhang et al., 2023; Tang et al., 2023; Lee et al., 2023) for consistent generation are usually employed in generating arbitrary-sized images or panoramas from smaller pieces. These methods involve concurrently generating these multiple pieces and merging their intermediate results in the overlapping areas within the sampling process. For example, MultiDiffusion (Bar-Tal et al., 2023) averages the diffusion model predictions to reconcile the different denoising processes. Recent work, TRF (Feng et al., 2024) extends this joint generation approach to the bounded video generation taking two end frames as input. By running two parallel image-to-video generations guided by the start and end frames, it merge their outputs

---
[1]https://www.pexels.com/

by averaging in each denoising step. However, a significant drawback of this method is that it cannot generate coherent motion in-between: simply fusing a forward video generation from the first end frame and the reversed forward video starting from the second end frame using a single image-to-video model designed for forward motion only causes the generated videos to oscillate between moving forward and then reversing, rather than continuously progress forward as our method does.

## 3 BACKGROUND

We introduce some background on Stable Video Diffusion (Blattmann et al., 2023a), the base image-to-video diffusion model used in our work, and then specifically explain the temporal self-attention layers within its architecture, which are key to modeling motion within the generated video.

### 3.1 STABLE VIDEO DIFFUSION

Diffusion models are trained to convert random noise into high-resolution images/videos via an iterative sampling process (Dhariwal & Nichol, 2021; Sohl-Dickstein et al., 2015; Song & Ermon, 2019; Song et al., 2020a;b; Liu et al., 2022). This sampling process aims to reverse a fixed, time-dependent destructive process (forward process) that gradually corrupts data by adding Gaussian noise. In particular, Stable Video Diffusion (SVD) is a latent diffusion model where the diffusion process operates in the latent space of a pre-trained autoencoder with encoder $\mathcal{E}(\cdot)$ and decoder $\mathcal{D}(\cdot)$.

In the forward process, a video sample $\mathbf{x} = \{I_0, I_1, ..., I_{N-1}\}$ composed of $N$ frames, is first encoded in the latent space $\mathbf{z} = \mathcal{E}(\mathbf{x})$, then the intermediate noisy video at time step $t$ is created as $\mathbf{z}_t = \alpha_t \mathbf{z} + \sigma_t \boldsymbol{\epsilon}$, where $\boldsymbol{\epsilon} \sim \mathcal{N}(\mathbf{0}, \mathbf{I})$ is Gaussian noise, and $\alpha_t$ and $\sigma_t$ define a fixed noise schedule. The denoising network $f_\theta$ receives this noisy video latent $\mathbf{z}_t$ and the conditioning $\mathbf{c}$ computed from the input image, i.e., the first frame $I_0$ in the video, and is trained by minimizing the loss:

$$\mathcal{L}(\theta) = \mathbb{E}_{t \sim U[1,T], \boldsymbol{\epsilon} \sim \mathcal{N}(\mathbf{0},\mathbf{I})}[\|f_\theta(\mathbf{z}_t; t, \mathbf{c}) - \mathbf{y}\|_2^2]$$

where the target vector $\mathbf{y}$ here is $\mathbf{v} = \alpha_t \boldsymbol{\epsilon} - \sigma_t \mathbf{z}_t$, referred to as v-prediction.

Once the denoising network is trained, starting from pure noise $\mathbf{z}_T \sim \mathcal{N}(\mathbf{0}, \mathbf{I})$, the sampling process iteratively denoises the noisy latent by predicting the noise in the input and then applying an update step to remove a portion of the estimated noise from the noisy latent

$$\mathbf{z}_{t-1} = \text{update}(\mathbf{z}_t, f_\theta(\mathbf{z}_t; t, \mathbf{c}); t)$$

until we get clean latent $\mathbf{z}_0$, followed by decoding $\mathcal{D}(\mathbf{z}_0)$ to get the generated video. The exact implementation of the $\text{update}(\cdot, \cdot)$ function depends on the specifics of the sampling method; SVD uses EDM sampler (Karras et al., 2022).

### 3.2 TEMPORAL SELF-ATTENTION

The denoising network $f_\theta$ in SVD is a 3D U-Net, composed of "down", "mid", and "up" blocks. Each block contains spatial layers interleaved with temporal layers, with the temporal self-attention layers responsible for modeling motion in the generated video. This layer takes a spatio-temporal tensor $X \in \mathbb{R}^{1 \times N \times H \times W \times C}$ as input, where $N$ is the number of frames, and $C$ is the number of channels. Here we use batch size of 1 for simplicity. The tensor is reshaped by moving the spatial dimensions $(H, W)$ into the batch dimension. This creates $X' \in \mathbb{R}^{HW \times N \times C}$, where self-attention operates solely on the temporal axis. More specifically, $X'$ is projected through three separate matrices $W_q, W_k, W_v \in \mathbb{R}^{d \times C}$ ($d$ is the dimensionality of the projected space.), resulting in the corresponding query ($Q = W_q X'$), key ($K = W_k X'$) and value ($V = W_v X'$) features. Then the scale-dot product attention is applied:

$$\text{Attention}(Q, K, V) = \text{softmax}(QK^T / \sqrt{d})V$$

The attention output is fed through another linear layer $W_o$ to get the final output. We refer to $A = QK^T \in \mathbb{R}^{HW \times N \times N}$ as the temporal self-attention map, which models the inter-frame correlations per spatial location. This temporal attention mechanism allows each frame's updated feature to gather information from other frames.

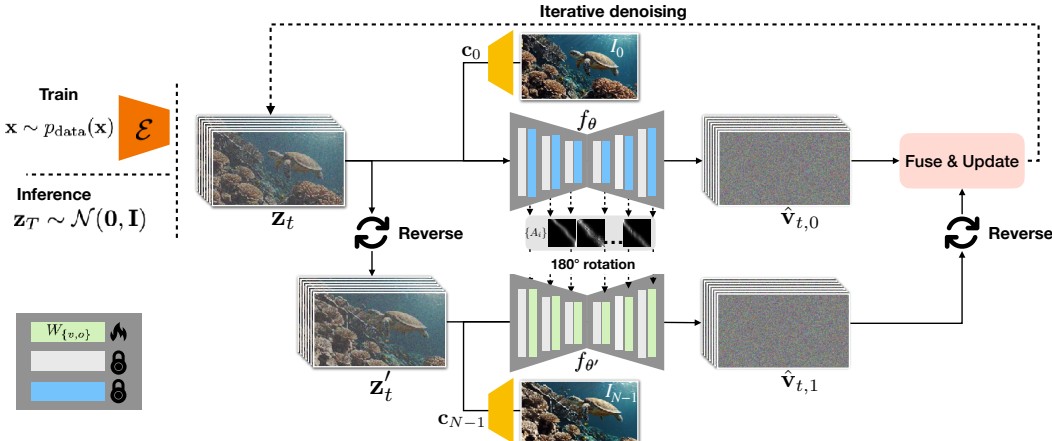

Figure 1: **Method overview.** In the lightweight backward motion fine-tuning stage, an input video $\mathbf{x} = \{I_0, I_1, ..., I_{N-1}\}$ is encoded into the latent space by $\mathcal{E}(\mathbf{x})$, and noise is added to create noisy latent $\mathbf{z}_t$; during inference, $\mathbf{z}_t$ is created by iterative denoising starting from $\mathbf{z}_T \sim \mathcal{N}(\mathbf{0}, \mathbf{I})$. (1) **Forward motion prediction:** we first take the conditioning $\mathbf{c}_0$ of the first input image (inference stage) or the first frame in the video (training stage) $I_0$, along with the noisy latent $\mathbf{z}_t$ to feed into the pre-trained 3D U-Net $f_\theta$ to get the noise predictions $\hat{\mathbf{v}}_{t,0}$, as well as the temporal self attention maps $\{A_i\}$. (2) **Backward motion prediction:** We reverse the noisy latent $\mathbf{z}_t$ along temporal axis to get $\mathbf{z}'_t$. Then we take the conditioning $\mathbf{c}_{N-1}$ of the second input image, or the last frame in the video $I_{N-1}$, along with the 180-degree rotated temporal self-attention maps $\{A'_i\}$, and feed them through the fine-tuned 3D U-Net $f_{\theta'}$ for backward motion prediction $\hat{\mathbf{v}}_{t,1}$. (3) **Fuse and update:** The predicted backward motion noise is reversed again to fuse with the forward motion noise to create consistent motion path. Note that only the value and output projection matrices $W_{\{v,o\}}$ in the temporal self-attention layers (**green**) are fine-tuned; see Fig. 2 for more details.

## 4 METHOD

Given a pair of keyframes $I_0$ and $I_{N-1}$, our goal is to generate a video $\{I_0, I_1, I_2, ...., I_{N-1}\}$ that begins with frame $I_0$ and ends with frame $I_{N-1}$, leveraging the pre-trained image-to-video Stable Video Diffusion (SVD) model. The generated video should exhibit a natural and consistent motion path, such as a car traveling or a person walking in a steady direction.

Image-to-video models typically generate video with motions that run forward in time. It is primarily the temporal self-attention layers that learn this motion-time association. In Sec 4.1, we discuss how this forward motion can be reversed by rotating the temporal self-attention maps by 180 degrees. Then we introduce an efficient lightweight fine-tuning technique to reverse this association and enable SVD to generate backward motion videos from the input image in Sec. 4.2. Finally we present our dual-directional sampling approach that fuses the forward motion generation starting with frame $I_0$ and backward motion video generation starting with frame $I_{N-1}$ in a consistent manner in Sec. 4.3. An overview of our method is shown in Fig. 1.

### 4.1 REVERSE MOTION-TIME ASSOCIATION BY SELF-ATTENTION MAP ROTATION

The temporal self-attention maps $\{A_i\}$ in the network $f_\theta$ feature the forward motion trajectory in video $\{I_0, I_1, ..., I_{N-1}\}$. By rotating these attention maps by 180 degrees, we obtain a new set $\{A'_i\}$ that depicts the opposite backward motion, corresponding to the reversed one $\{I_{N-1}, I_{N-2}, ..., I_0\}$ starting from the last frame $I_{N-1}$.

Specifically, rotating the temporal self-attention maps by 180 degrees—flipping them vertically and horizontally—yields a backward motion opposite to the original forward motion. For example, consider attention map $A$; the rotated map $A'_{N-j,N-k} = A_{j,k}$, where $A_{j,k}$ indicates the attention score between the j-th and k-th frames ($I_j$ and $I_k$). In the corresponding reversed video, the reverse frame indices $N - j$ and $N - k$ maintain the same relative response.

## 4.2 LIGHTWEIGHT BACKWARD MOTION FINE-TUNING

We introduce a lightweight fine-tuning framework that specifically fine-tunes the value and output projection matrix $W_v, W_o$ in the temporal self-attention layers, using the 180-degree rotated attention map from the forward video as additional input (see Fig. 2). We use $f_{\theta'}(\mathbf{z}_t; t, \mathbf{c}, \{A'_i\})$ to denote the backward motion generation network. This fine-tuning approach offers two key advantages:

First, by utilizing existing forward motion statistics from the pre-trained SVD model, fine-tuning $W_{\{v,o\}}$ simplifies the model's task to focus on learning how to synthesize reasonable content when operating in reverse. This strategy requires significantly less data and fewer parameters compared to full model fine-tuning. Second, it enables the control for the model to generate a backward motion trajectory corresponding to the opposite of the forward trajectory described by the attention map. This feature is particularly beneficial when planning to merge forward and backward motions converging towards each other, and thus achieving forward-backward consistency.

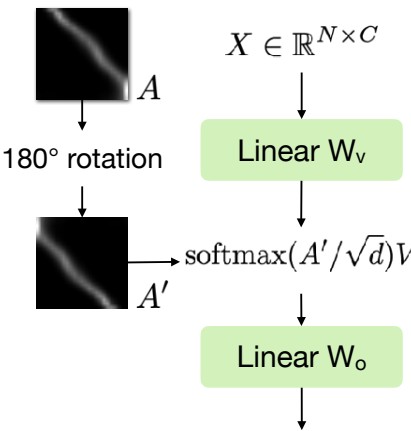

Figure 2: **Temporal self-attention module in the backward motion generation.** Given input tensor $X$, our attention mechanism additionally takes the respective attention map $A$ from the pre-trained SVD featuring forward motion, rotating it by 180 degrees to create a reverse motion-time association $A'$. Note that $W_{\{v,o\}}$ are the only trainable parameters in this module.

The detailed training process is shown in Alg. 1. For latent video $\mathbf{z} \in \mathbb{R}^{1 \times N \times C \times H \times W}$, we denote flip(z) specifically by the second dimension, i.e., reversing the latent video along the time axis. In every training iteration, we sample an input video of $N$ frames, and random time step $t$, then the noisy video latent $\mathbf{z}_t$ is created by adding the noise in that time step. The noisy video latent along with the input conditioning $\mathbf{c}_0$ (computed from $I_0$) is fed into the pre-trained 3D U-Net $f_\theta$ to extract the self attention maps $\{A_i\}$ from the temporal attention layers. Then we reverse the noisy video latent, along with the last frame conditioning $\mathbf{c}_{N-1}$, feed them into the backward motion 3D-U-Net $f_{\theta'}$. The loss function is computed by taking the predictions of the network and the ground truth reverse video.

---

**ALGORITHM 1:** Light-weight backward motion fine-tuning

---

**Input:** $f_\theta, p_{\text{data}}(\mathbf{x}), \mathcal{E}(\cdot)$
**while** *not converged* **do**

    Sample $\mathbf{x} \sim p_{\text{data}}(\mathbf{x}), \mathbf{x} = \{I_n\}_{n=0}^{N-1}, \mathbf{z} = \mathcal{E}(\mathbf{x})$;
    Compute conditioning $\mathbf{c}_0$ from $I_0$;
    $t \sim \text{Uniform}(\{1, ..., T\}), \boldsymbol{\epsilon} \sim \mathcal{N}(\mathbf{0}, \mathbf{I})$;
    $\mathbf{z}_t = \alpha_t \mathbf{z} + \sigma_t \boldsymbol{\epsilon}$;
    $\{A_i\} = \text{extract\_attention\_map}(f_\theta(\mathbf{z}_t; t, \mathbf{c}_0))$ ;
    $\mathbf{z}'_t = \text{flip}(\mathbf{z}_t)$;
    Compute conditioning $\mathbf{c}_{N-1}$ from $I_{N-1}$;
    Take gradient descent step on $\nabla_{W_{\{v,o\}}} \|f_{\theta'}(\mathbf{z}'_t; t, \mathbf{c}_{N-1}, \{A'_i\}) - \mathbf{y}\|_2^2, \mathbf{y} = \alpha_t \text{flip}(\boldsymbol{\epsilon}) - \sigma_t \mathbf{z}'_t$;

**end**
**Return:** $W_{\{v,o\}}$

---

## 4.3 DUAL-DIRECTIONAL SAMPLING
### WITH FORWARD-BACKWARD MOTION CONSISTENCY

Our complete *dual-directional sampling* process is detailed in Alg. 2. Given a pair of keyframes $I_0$ and $I_{N-1}$, their corresponding conditioning $\mathbf{c}_0$ and $\mathbf{c}_{N-1}$ are pre-computed. Then each sampling step (illustrated in Figure 1) works as follows:

(1) Forward motion denoising with $I_0$ as input: The noisy video latent $z_t$ along with the conditioning $\mathbf{c}_0$ is fed into the pre-trained 3D U-Net $f_\theta$ in SVD to predict the noise volume $\hat{\mathbf{v}}_{t,0}$. Additionally, the temporal self-attention maps $\{A_i\}$ in the 3D U-Net are extracted.

(2) Backward motion denoising with $I_{N-1}$ as input: The noisy video $\mathbf{z}_t$ is flipped along the temporal dimension to create the reverse video latent $\mathbf{z}'_t$ corresponding to the backward motion. This backward video, along with the conditioning $\mathbf{c}_{N-1}$, as well as the 180-degree rotated attention maps $\{A'_i\}$, are fed into our fine-tuned 3D U-Net $f_{\theta'}$. This step predict the noise volume $\hat{\mathbf{v}}_{t,1}$ representing a reverse motion from $I_{N-1}$.

(3) Finally, the predicted noise volumes from both forward and reverse motion paths are fused and then denoised using the update$(\cdot, \cdot)$ function to create less noisy video $\mathbf{z}_{t-1}$. In this way, we ensure forward-backward consistency and thus a consistent moving direction in the generated video. The fuse$(\cdot, \cdot)$ function performs a simple average. In practice, we also adopt per-step recurrence to enhance the fusion as seen in (Bansal et al., 2023; Feng et al., 2024), by re-injecting Gaussian noise into the update $\mathbf{z}_{t-1}$ and repeating the denoising 5 times before continuing the sampling for the next step.

---

**ALGORITHM 2:** Dual-directional diffusion sampling

**Input:** $I_0, I_{N-1}, f_\theta, f_{\theta'}, \mathcal{D}(\cdot)$
Compute condition $\mathbf{c}_0, \mathbf{c}_{N-1}$ from $I_0, I_{N-1}$;
Set $\mathbf{z}_T \sim \mathcal{N}(\mathbf{0}, \mathbf{I})$;
**for** $t \leftarrow T$ **to** $1$ **do**
$\quad \hat{\mathbf{v}}_{t,0} = f_\theta(\mathbf{z}_t; t, \mathbf{c}_0)$;
$\quad \{A_i\} = \text{extract\_attention\_map}(f_\theta(\mathbf{z}_t; t, \mathbf{c}_0))$;
$\quad \mathbf{z}'_t = \text{flip}(\mathbf{z}_t)$;
$\quad \hat{\mathbf{v}}_{t,1} = f_{\theta'}(\mathbf{z}'_t; t, \mathbf{c}_{N-1}, \{A'_i\})$ ;
$\quad \hat{\mathbf{v}}'_{t,1} = \text{flip}(\hat{\mathbf{v}}_{t,1})$ ;
$\quad \hat{\mathbf{v}}_t = \text{fuse}(\hat{\mathbf{v}}_{t,0}, \hat{\mathbf{v}}'_{t,1})$;
$\quad \mathbf{z}_{t-1} = \text{update}(\mathbf{z}_t, \hat{\mathbf{v}}_t; t)$
**end**
**Return:** $\mathcal{D}(\mathbf{z}_0)$

---

### 4.4 IMPLEMENTATION DETAILS

Our lightweight fine-tuning technique fine-tunes less than $2\%$ of the U-Net parameters, and does not rely on large collection of training videos. So we collected $100$ high quality videos which are originally generated from SVD from a community website[2] as our training data. Our experimental results show that our method generalizes well to the real image data. We select the ones with large object motion such as animal running, vehicle moving, people walking, and so on. We use the Adam optimizer with learning rate of $1e-4$, $\beta_1 = 0.9, \beta_2 = 0.999$, and weight decay of $1e-2$. The training takes around $15K$ iterations with batch size of $4$. We trained on $4$ A100 GPUs. For sampling, we apply $50$ sampling steps. For other parameters in SVD, we use the default values: *motion bucket id* $= 127$, *noise aug strength* $= 0.02$.

## 5 EXPERIMENTS

In Figs. 3, 4, 5, we demonstrate that our approach successfully generates high quality videos with consistent motion given distant keyframes. We highly recommend viewing the videos in the supplementary to see the results more clearly. Sec. 5.1 describes the data we used to evaluate our method and the baselines. Sec. 5.2 demonstrates how our method outperforms traditional frame interpolation method FILM, and the recent work TRF (Feng et al., 2024) that also leverages SVD for video generation. Sec. 5.3 justifies our design decisions with an ablation study. Sec. 5.4 discusses the optimal scenarios where our method excels and sub-optimal ones where it outperforms the baselines but remains limited by SVD itself. Sec. 5.5 discusses our failure cases.

### 5.1 EVALUATION DATASET

We use two high-resolution (1080p) datasets for evaluations: (1) The Davis dataset (Pont-Tuset et al., 2017), where we create a total of $117$ input pairs from all of the videos. This dataset mostly features

---

[2]https://www.stablevideo.com/

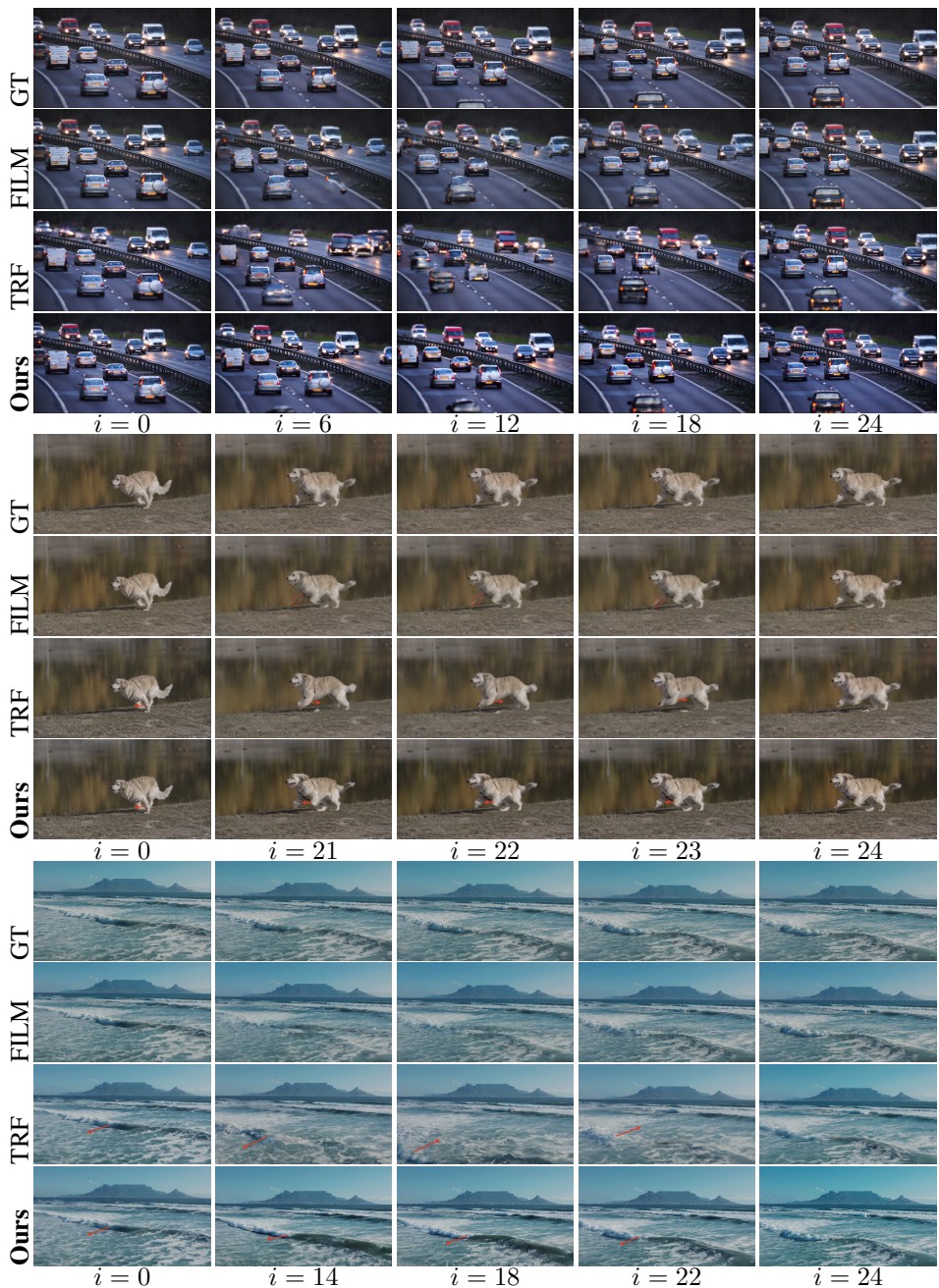

Figure 3: **Qualitative baseline comparisons.** Leftmost ($i = 0$) and rightmost columns ($i = 24$): start and end frames. TRF generates back-and-forth motions, such as vehicles moving forward and then reversing. FILM struggles to find correspondences when the input frames are distant and morphs from the first frame to the last. The red arrow indicates the direction of motion. We recommend viewing the supplementary videos.

subject articulated motions, such as animal or human motions. (2) The Pexels dataset, where we collect a total of 106 input keyframe pairs from a compiled collection of high resolution videos on Pexels[3], featuring directional dynamic scene motions such as vehicles moving, animals, or people running, surfing, wave movements, and time-lapse videos. All input pairs are at least 25 frames apart and have the corresponding ground truth video clips.

---

[3]https://www.pexels.com/

## 5.2 BASELINE COMPARISONS

We mainly compare our approach to FILM (Reda et al., 2022), the current state-of-the-art frame interpolation method for large motion, and TRF (Feng et al., 2024) which also adapts SVD for bounded generation. We show representative qualitative results in Figs. 3, 5. In addition, we also include results for the keyframe interpolation feature from the recent work DynamiCrafter (Xing et al., 2023)—a large-scale image-to-video model. The keyframing feature is modified from it and specially trained to accept two end frames as conditions, while we focus on how to *adapt* a pretrained image-to-video model in a lightweight way with small collection of training videos and much less computational resources. This feature generates videos at resolution $512 \times 320$, while ours generates at resolution $1024 \times 576$. Nonetheless, we present its results for reference.

**Quantitative evaluation** For each dataset, we evaluate the generated in-between videos using FID (Heusel et al., 2017) and FVD (Ge et al., 2024), widely used metrics for evaluating generative models. These two metrics measure the distance between the distributions of generated frames/videos and actual ones. The results are shown in Tab. 1, and our method outperforms **all** of the baselines by a significant margin.

|  | Pexels | | Davis | |
|---|---|---|---|---|
|  | FID ↓ | FVD ↓ | FID ↓ | FVD↓ |
| FILM (Reda et al., 2019) | 25.16 | 371.83 | 41.85 | 1048.65 |
| TRF (Feng et al., 2024) | 31.43 | 563.16 | 36.79 | 563.07 |
| DynamiCrafter (Xing et al., 2023) | 32.06 | 393.12 | 38.32 | 439.74 |
| Ours w/o RA | 26.42 | 458.76 | 36.70 | 549.98 |
| Ours w/o FT | 37.68 | 555.10 | 47.23 | 604.76 |
| **Ours** | **22.99** | **306.84** | **32.68** | **424.69** |

Table 1: Comparisons with baselines and our ablation variants. *Ours w/o RA*: full pipeline with fine-tuning all parameters $W_{\{q,k,v,o\}}$ without using the 180-degree rotated temporal attention map. *Ours w/o FT*: full pipeline using rotated attention map only in the "up" blocks and without fine-tuning $W_{\{v,o\}}$ for backward motion.

**Comparison to FILM** The flow-based frame interpolation method FILM suffers from two problems. First, it struggles to find correspondences in scenes with large motions. For example, in the second row of Fig. 3, in a highway where vehicles moving in both directions, FILM fails to find the correspondence between the moving cars across the input keyframes, resulting in implausible intermediate motions. For example, some cars in the first frame disappear in the middle and reappear at the end. Second, it generates undesirable unambiguous motion which takes the shortest path between the end frames. In the example in Fig. 5, given two similar-looking frames that captures different states of a person running, FILM produces a motion that merely translates the person across the frames, losing the natural kinematic motions of the legs.

**Comparison to TRF** TRF fuses the forward video generation starting from the first frame and the reversed forward video starting from the second frame, both using the original SVD. The reversed forward video from the second frame creates a backward motion video that ends at the second frame. Fusing these generation paths results in a back-and-forth motion in the generated videos. One notable effect we observe with TRF is that the generated videos exhibit a pattern of progressing forward first and then reversing to the end frame. For example, in the third row of Fig. 3, we can see the red truck moving backward over time; in the seventh row, the dog's legs are moving backwards, leading to unnatural motions. In contrast, our approach fine-tunes SVD to generate a backward video starting from the second frame in the opposite direction to the forward video from the first frame.This forward-backward motion consistency leads to the generation of a motion-consistent video.

## 5.3 ABLATIONS

In Fig. 4 and Tab. 1, we show visual and quantitative comparisons to simpler versions of our method to evaluate the effect of the key components in our method.

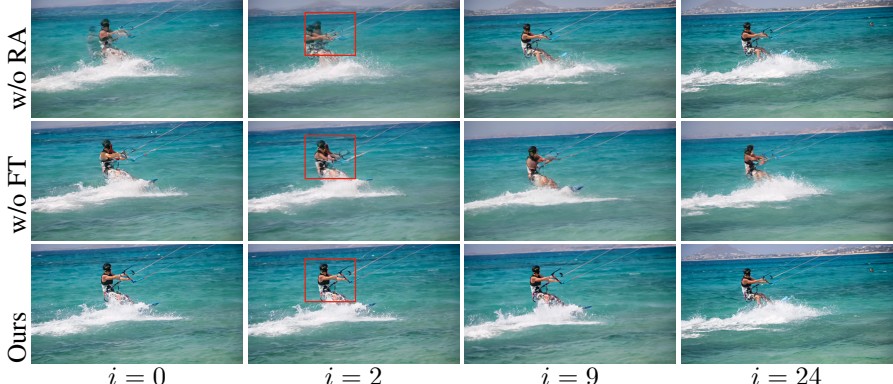

Figure 4: **Ablation study.** We evaluate other options for generating in-between motion consistency. (1) *Ours w/o RA*: full pipeline with fine-tuning all parameters $W_{\{q,k,v,o\}}$ in the temporal attention layers but without using 180-degree rotated temporal self-attention maps as extra input (top row). (2) *Ours w/o FT*: full pipeline without fine-tuning $W_{\{v,o\}}$ for backward motion (second row). The differences are highlighted in the red rectangle.

**Fine-tuning without rotated attention map (Ours w/o RA)** We compare with a variant that fine-tunes all parameters in the temporal self-attention layers, namely, $W_{\{q,k,v,o\}}$, but without using the 180-degree rotated temporal self-attention map from the forward video as extra input. Though fine-tuning all parameters can generate backward motion from the second input image, there is no guarantee that the backward motion will mirror the forward motion from the first input image. This discrepancy makes it hard for the model to reconcile the two motion paths, often resulting in blending artifacts, as shown in the top row of Fig. 4. In contrast, fine-tuning $W_{\{v,o\}}$ with the rotated attention maps generates coherent and high-fidelity in-between videos.

**Fine-tuning $W_{\{v,o\}}$ vs. no fine-tuning (Ours w/o FT)** In Sec. 4.1, we show that rotating the temporal attention maps by 180 degrees reverses the motion-time association, creating a backward motion trajectory. Here we show that fine-tuning the value and output projection matrices $W_{v,o}$ is necessary for the model to synthesize high-fidelity content given the input backward motion-time association. We run our full pipeline without any fine-tuning, and our attention map rotation operation is only applied to the "up" blocks in this variant. As shown in the second row of Fig. 4 and Tab. 1, without fine-tuning these parameters, the model can create consistent motion but suffers from poor frame quality due to the low frame quality of the backward video generation. For example, the person is disfigured in the generated video. Note that applying the attention map rotation operation to the "down" and "mid" blocks in this variant worsens visual fidelity even further; thus, we show the best-case scenario without fine-tuning (i.e., applying rotated attention maps to the "up" blocks only).

## 5.4 Optimal and Sub-optimal Scenarios

Our method is limited by the motion quality and priors learned by SVD. Firstly, our empirical experiments indicate that SVD works well with generating rigid motions, but struggles with non-rigid, articulated movements. It has difficulty accurately rendering the limb movements of animal/people. In Fig. 5, though our method significantly improves upon FILM and TRF, it still appears unnatural compared to the ground truth movements. The bottom row, showing the sequence generated by SVD using only the first input frame, confirms that SVD itself struggles to generate natural running movements in between.

## 5.5 Failures

When the input pairs are captured at such distant intervals that they have sparse correspondences, as shown in Fig. 6, where only a small portion of cars appear in both input frames, it becomes difficult for our method to fuse the forward and backward motions. This situation, where the overlapping areas are minimal, leads to artifacts in the intermediate frames.

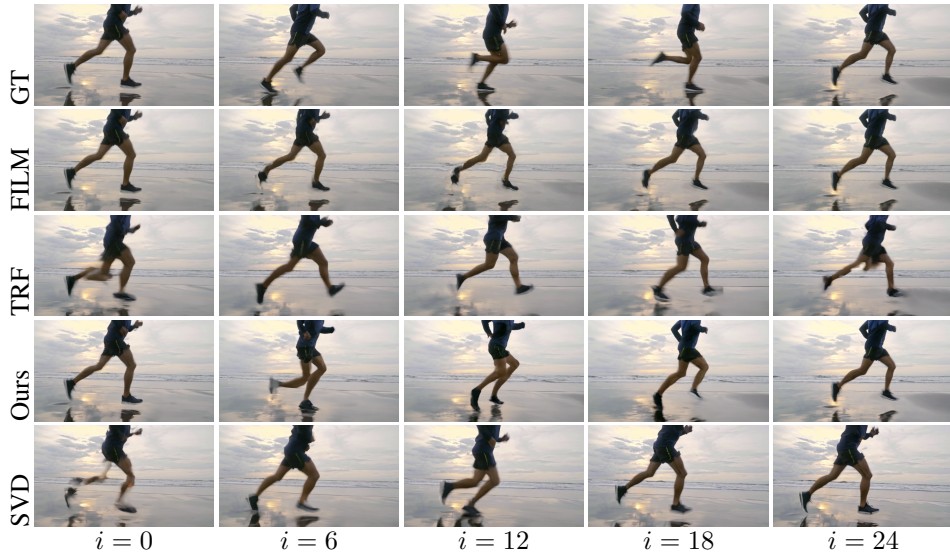

Figure 5: Our method outperforms FILM and TRF in generating articulated movements inbetween, but still struggles to create natural kinematic motions because of the limitation of SVD itself failing to generated complex kinematics (bottom row). Note that the input image serve as conditioning to SVD, so generated first frame might differ from the input image if SVD struggles to create plausible videos from that input.

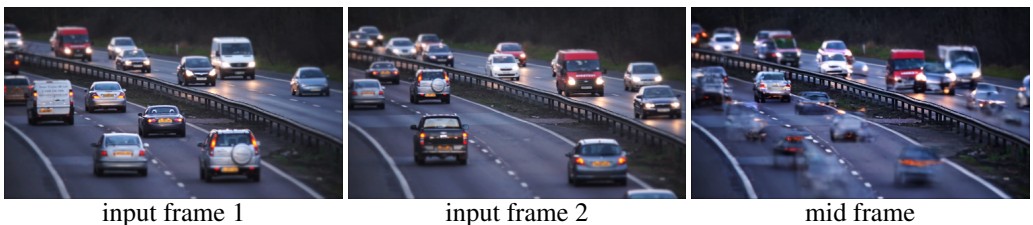

Figure 6: **Failure case.** Our method fails to work well in the cases where input pairs have sparse correspondences.

## 6 DISCUSSIONS & LIMITATIONS

Our method is limited by the motion quality of the underlying base model, Stable Video Diffusion (SVD), as discussed in Sec. 5.4. Another limitation is that SVD has strong motion priors derived from the input image, tending to generate only specific motions for a given input. As a result, the actual motion required to connect the input key frames may not be represented within SVD's motion space, making it challenging to synthesize plausible intermediate videos. However, with advancements in large scale image-to-video models like SoRA[4], we are optimistic that these limitations can be addressed in the future. Including better motion datasets and incorporating articulated motion/physical movement priors may also help. Another potential improvement involves using motion heuristics between the input key frames to prompt the image-to-video model to generate more accurate in-between motions.

**Acknowledgements** We thank Meng-Li Shih, Bowei Chen, and Dor Verbin for helpful discussions and feedback. This work was supported by the UW Reality Lab and Google.

---

[4]https://openai.com/index/sora/

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

# A APPENDIX

## A.1 SENSITIVITY TO THE SCALE OF TRAINING SET

As stated in Sec. 4.4, our method fine-tunes fewer than $2\%$ parameters of the original model by using the attention map from the pretrained model, and thus we reduce the need for extensive training data. We use 100 synthetic training videos in our experiments. Here we we conduct an ablation by varying the training dataset size to be 50 and 150 videos, and evaluate the performance as done in Tab. 1. Our method still outperforms the baselines even with a training size of 50, and its performance increases slowly as more data is added (see Fig. 7).

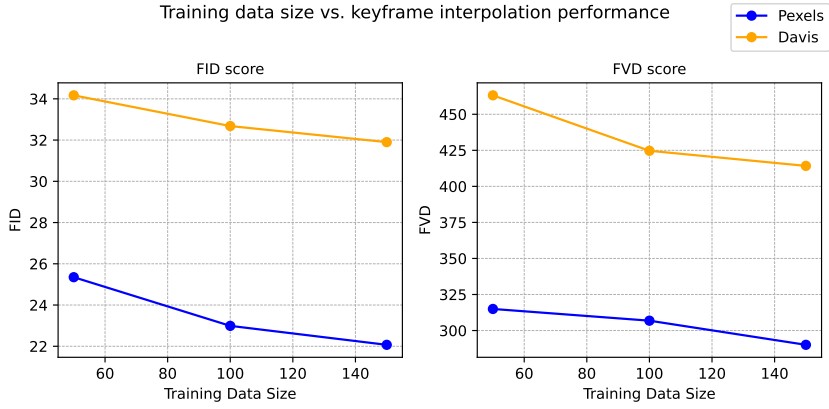

Figure 7: Ablation on how the scales of the training dataset affect our model's performance.

|  | **Pexels** | | **Davis** | |
|---|---|---|---|---|
|  | FID $\downarrow$ | FVD $\downarrow$ | FID $\downarrow$ | FVD $\downarrow$ |
| FILM | 25.16 | 371.83 | 41.85 | 1048.65 |
| TRF | 31.43 | 563.16 | 36.79 | 563.07 |
| Ours ($motion\ bucket\ id = 255$) | 22.18 | 306.61 | 34.06 | 426.53 |
| Ours ($motion\ bucket\ id = 65$) | 23.33 | 270.27 | 32.75 | 491.85 |
| Ours ($motion\ bucket\ id = 127$) | 22.99 | 306.84 | 32.68 | 424.69 |

Table 2: Ablation on how the conditioning parameters $motion\ bucket\ id$ in Stable Video Diffusion affect our model's performance. Our method uses 127 as default in the paper.

## A.2 THE INFLUENCE OF MOTION BUCKET ID IN STABLE VIDEO DIFFUSION

Stable Video Diffusion[5] (Blattmann et al., 2023a) takes $motion\ bucket\ id$ as micro conditioning parameter in the video generation process, which is expected to affect the motion magnitude the generated video: higher values result in more dynamic video and vice versa. However, keyframe interpolation is a more constrained task where the second end frame provides additional guidance for the generation (Feng et al., 2024). Here we experiment with different $motion\ bucket\ id$ values in Tab. 2, and our method still outperforms TRF and FILM. On the Pexels dataset, $motion\ bucket\ id$ of 65 results in better FVD score (generated motion closer to the ground truth videos), However, the same value results in worse FVD score on the Davis dataset. This discrepancy is likely due to motion difference between the two datasets.

## A.3 PSEUDO CODE FOR LIGHTWEIGHT BACKWARD MOTION FINE-TUNING AND DUAL DIRECTIONAL SAMPLING

---

[5]We use the public available model weights `https://huggingface.co/stabilityai/stable-video-diffusion-img2vid-xt`

```python
def get_trainable_params(rev_UNet):
    # Only finetune Wv and Wv in temporal self attention layers
    rev_unet_train_params = []
    for name, param in rev_UNet.named_parameters():
        if 'temporal_transformer_blocks.0.attn1.to_v.weight' in name
        or 'temporal_transformer_blocks.0.attn1.to_out.0.weight' in name:
            rev_unet_train_params.append(param)
            param.requires_grad = True
    return rev_unet_train_params

# Backward motion fine-tuning
rev_UNet = copy.deepcopy(ori_UNet)
ori_UNet.requires_grad(False) # pretrained 3DUNet in SVD
rev_UNet.requires_grad(False) # backward motion UNet to be fine-tuned
optimizer = optim.AdamW(get_trainable_params(rev_UNet)
for epoch in range(0, num_train_epochs):
    rev_UNet.train()
    loss = 0.
    for batch_video in train_dataloader:
        I_0 = batch_video[:, 0] # get the first frame from the video
        I_N = batch_video[:, -1] # get the last frame from the video
        c_0 = compute_image_conditioning(I_0)
        c_N = compute_image_conditioning(I_N)

        batch_video = rearrange(batch_video, "b f c h w -> (b f) c h w")
        z = vae.encode(batch_video)
        z = rearrange(z_0, "(b f) c h w -> b f c h w", f=num_frames)
        z_rev = torch.flip(z, dims=(1,)) # GT reverse latent video

        noise = torch.rand_like(z)
        t = torch.randint(0, num_train_timesteps)
        z_t = noise_scheduler.add_noise(z, noise, t)
        z_t_rev = torch.flip(z_t, dims=(1,))

        pred_noise_1, attention_maps = ori_UNet(z_t, t, c_0)

        # rotate attention maps by 180 degree
        rotated_attention_maps = [torch.flip(attention_map, dims=(-2, -1))
                    for attention_map in attention_maps]

        pred_noise_2 = rev_UNet(z_t_rev, t, c_N, rotated_attention_maps)
        pred_z_rev = noise_scheduler.predict_denoised_sample(pred_noise_2)
        loss += mse_loss(pred_z_rev, z_rev)
    loss.backward()
    optimizer.step()
    optimizer.zero_grad()
return rev_UNet
```

Figure 8: Pytorch pseudocode for lightweight backward motion fine-tuning.

```python
# Dual directional diffusion sampling for generating in-between video
# ori_UNet: pretrained 3DUNet in SVD
# rev_UNet: fine-tuned backward motion UNet

c_0 = compute_image_conditioning(I_0)
c_N = compute_image_conditioning(I_N)
z_t = torch.randn(latent_shape) # initialize video latent variable
timesteps = scheduler.set_timesteps(num_steps)
for i, t in enumerate(timesteps):
    # predicted noise in forward motion start from I_0
    pred_noise_1, attention_maps = ori_UNet(z_t, t, c_0)
    if do_classifier_free_guidance:
        pred_noise_uncond_1 = ori_UNet(z_t, t, null_conditioning)
        pred_noise_1 =  pred_noise_uncond_1 +
        guidance_scale * (pred_noise_1 - pred_noise_uncond_1)

    rotated_attention_maps = [torch.flip(attention_map, dims=(-2, -1))
                        for attention_map in attention_maps]

    # predicted noise in backward motion start from I_N
    z_t_rev = torch.flip(z_t, dims=(1,))
    pred_noise_2 = rev_UNet(z_t_rev, t, c_N, rotated_attention_maps)
    if do_classifier_free_guidance:
        pred_noise_uncond_2 = rev_UNet(z_t_rev, t, null_conditioning,
                                rotated_attention_maps)
        pred_noise_2 = pred_noise_uncond_2
        + guidance_scale * (pred_noise_2 - pred_noise_uncond_2)

    pred_noise_2 = torch.flip(pred_noise_2, dims=(1,))

    pred_noise = (pred_noise_1 + pred_noise)/2. # fuse
    z_t = scheduler.update(z_t, pred_noise, t) # denoise
output_video = vae.decode(z_t)
return output_video
```

Figure 9: Pytorch pseudocode for dual directional diffusion sampling.

