# OpenReview forum: "Generative Inbetweening: Adapting Image-to-Video Models for Keyframe Interpolation"
_ICLR.cc/2025/Conference — ICLR 2025 Poster_

### Official Review · Reviewer_MnYK · 2024-10-21

**Soundness:** 3
**Presentation:** 3
**Contribution:** 3
**Rating:** 6
**Confidence:** 4

**Summary:**

This paper addresses the keyframe interpolation problem by leveraging the large-scale image-to-video diffusion model, Stable Video Diffusion, to generate frames between a pair of keyframes.

Unlike traditional image-to-video models that generate frames in a forward-moving manner, this paper proposes finetuning the model for backward-moving videos and utilizing both the original and finetuned models together during inference.

To leverage the knowledge from the forward-moving model, only the value and output projection matrices of the 3D self-attention layers are trained, and the attention maps from the forward-moving videos are rotated by 180 degrees and inserted into the finetuned backward-moving model.

During inference, the attention maps generated by the forward-moving model are rotated and applied to the finetuned backward-moving model, and the predictions from both models are fused.

This approach demonstrates superior performance over FILM and TRF on the Davis and Pexels datasets, despite being trained on only 100 videos.

**Strengths:**

- This method is both parameter-efficient and data-efficient, making it highly effective even with limited resources.

- It leverages an open-source model, which enhances its accessibility and contributes to the broader video interpolation research community.

- It demonstrates superior qualitative and quantitative performance compared to FILM, a well-known method for large motion interpolation, as well as TRF, which also uses Stable Video Diffusion.

- In Section 5, the qualitative results are thoroughly explained, clearly highlighting the strengths of this approach in various aspects.

**Weaknesses:**

- Video interpolation performance can vary significantly based on FPS and the magnitude of motion, but the paper does not provide any analysis of these factors. Besides the motion bucket ID mentioned in the paper, Stable Video Diffusion also takes FPS as a condition. The paper would benefit from demonstrating whether the method still outperforms FILM and TRF when varying the motion bucket ID and FPS during finetuning and inference.

- The proposed method requires both the base forward-moving model and the finetuned backward-moving model during both training and inference, making it more computationally intensive compared to a baseline of
 fine-tuning on video interpolation dataset.

**Questions:**

- There are three publicly available weights for Stable Video Diffusion (img2vid, img2vid-xt, img2vid-xt-1-1). Which of these weights did the authors use?

- Stable Video Diffusion applies different classifier-free guidance scales to each frame. Did the authors use the same approach in this paper?

---

> ### Author Response · Authors · 2024-11-26
>
> > The influence of motion bucket id and FPS
>
> Though the micro-conditioning parameter motion bucket id in SVD might influence the motion magnitude in the generated video from a single image, keyframe interpolation is a more constrained task where the second end frame provides additional guidance for the generation.
>
> We experiment with motion bucket id values of  65 and 255, compared to the default value of 127 used in the paper,  and our method still outperforms TRF and FILM. On the Pexels dataset, a motion bucket ID of 65 results in better FVD score (generated motion closer to the ground truth videos), However, the same value results in worse FVD score on the Davis dataset. This discrepancy is likely due to motion difference between the two datasets. We have included these results in Table 2 and the discussion in the appendix in the revision.
>
> For the FPS parameter, we experiment with a much higher conditioning value of 25 for both our method and TRF, resulting in suboptimal results on Davis. Our method achieved an FVD score of 647.31, while TRF got 801.77, compared to their respective scores of  424.69 and 563.07 with the default FPS value of 7.  A large FPS value leads the SVD to generate suboptimal results and is not recommended.
>
> We agree that an interesting direction for future exploration is incorporating the motion direction and magnitude difference between keyframes as a prior to the model to achieve the optimal result per instance.
>
> > Computation efficiency
>
> Yes, we acknowledge that sampling using our method can be slower than a model fine-tuned directly for the interpolation task (since it requires passing through both the forward and backward motion networks), but the advantage of our approach lies in training: i.e., the ability to easily adapt a pretrained image-to-video model with minimal compute and training data.
>
> > Stable Video Diffusion model weights version
>
> We use img2vid-xt model weights, and we have included this detail in the revision.
>
> > Per-frame classifier free guidance
>
> In our case, we apply the same guidance scale 2.0 for each frame and a naive averaging weight of 0.5 for every frame during the fusion stage, rather than using a linearly increasing guidance scale ranging from 1.0 to 3.0 per frame as done in the original SVD model.

---

> > ### Comment · Reviewer_MnYK · 2024-11-27
> >
> > I appreciate that the rebuttal addresses my concerns, especially on micro-conditioning parameters. However, I partially disagree regarding the computational cost. Since the proposed method requires inference of a pre-trained model during fine-tuning, the statement "adapt a pre-trained image-to-video model with minimal compute" could be misleading.

---

> > > ### Author Response · Authors · 2024-11-27
> > >
> > > Thanks for the response. More precisely, we mean to say that our adaptation is data efficient and parameter efficient.

---

### Official Review · Reviewer_VhZE · 2024-10-31

**Soundness:** 3
**Presentation:** 3
**Contribution:** 3
**Rating:** 6
**Confidence:** 4

**Summary:**

The author proposes a novel method for distant keyframe interpolation, leveraging pretrained image-to-video diffusion models. This approach generates intermediate frames by predicting and fusing forward and backward video sequences, conditioned respectively on the given start and end frames. The author introduces a lightweight fine-tuning technique to tackle the key challenge of predicting backward video sequences from the end frame. Additionally, a dual-directional diffusion sampling strategy is employed to effectively fuse noise in both forward and backward directions.

**Strengths:**

- The paper is clear and easy to understand, with well-presented motivation and methodology.
- The proposed method is novel, straightforward, and effective, demonstrating improvements over the selected baseline interpolation methods.

**Weaknesses:**

Further ablation studies on the proposed method could explore:
1. **Training Dataset Scale**: In the paper, the model is fine-tuned with only 100 videos. It would be interesting to investigate how the scale of the training dataset affects the model’s performance.
2. **Fine-tuning Modules**: The paper fine-tunes only the value and output projection matrices in the self-attention layers of the backward framework. Since there might be a gap for the forward motion in the context of the image-to-video task and the interpolation task, it would be worth exploring whether the interpolation performance could be further improved by fine-tuning both the forward and backward framework matrices while preserving the attention map rotation operation.

**Questions:**

Please kindly refer to the Weaknesses.

---

> ### Author Response · Authors · 2024-11-26
>
> > How does the scale of the training dataset affect the model’s performance?
>
> By finetuning fewer than 2% parameters of the original model using the attention map from the pretrained model, we reduce the need for extensive training data. We conducted an additional ablation by varying the training dataset size between 50 and 150 videos, and evaluated the performance. Please see Figure 7 in the appendix in the submitted revision: our method outperforms the baselines even with a training size of 50, and its performance increases slowly as more data is added.
>
> > Fine-tuning modules
>
> We agree that fine tuning more parameters may further improve results, and is an interesting avenue for future work.

---

### Official Review · Reviewer_aMmk · 2024-10-31

**Soundness:** 4
**Presentation:** 3
**Contribution:** 3
**Rating:** 6
**Confidence:** 4

**Summary:**

The paper focuses on the keyframe interpolation problem, which has been overlooked in existing large-scale video generation models. The article proposes a solution to this task by treating keyframe interpolation as a forward video generation from the first frame and a backward generation from the last frame, followed by a coherent fusion of the generated frames. Based on this, the paper reuses existing large-scale image-to-video models to obtain a video generation model for backward motion by reversing temporal interactions. Additionally, it uses sampling techniques to blend paired frames generated by the forward and backward temporal directions with synchronized paths, producing intermediate frames.

**Strengths:**

This paper fills a gap in the field of large-scale video generation, specifically keyframe interpolation, at a related lower cost. As summarized earlier, this paper presents a novel pipeline to generate synchronized frames and targeted frame fusion techniques to achieve smooth transitional videos.

**Weaknesses:**

The keyframes shown in the paper have relatively small motion ranges and require extensive pixel mapping; otherwise, obvious artifacts occur (as mentioned in the limitations), making this approach unsuitable for large-scale object or camera movements.

**Questions:**

1. As noted in the weaknesses, I observed that in the cases provided, the camera and object movements between keyframes are slight. Can this method still perform well when there is a significant difference between the given keyframes? Additionally, when the keyframe difference is large, the backward generation may be unable to reuse the rotated attention matrix from the forward generation, potentially causing large discrepancies in frames generated at the same time step. In such cases, can fusion still work effectively?
2. This generation pipeline seems to require a substantial number of corresponding points between keyframes. Beyond the issue of low overlap mentioned earlier, I’m also curious whether the method could still generate smooth transitions if, for example, one object in the keyframes—such as a fish in the ocean—undergoes a mirrored flip, meaning every point has a mapped counterpart but with an orientation change.
3. The paper adopts simple averaging for intermediate frame fusion (line 281), but intuitively, frames generated closer to the initial keyframe might exhibit higher quality. Why not use weighted averaging instead? For example, linearly blending frames based on their proportional distance from each keyframe might yield smoother transitions and higher quality.

---

> ### Author Response · Authors · 2024-11-26
>
> > Can this method still perform well when there is a significant difference between the given keyframes?
>
> Performance drops with much larger differences between keyframes for a couple reasons: first, we use Stable Video Diffusion, which can generate only 25 frames, and struggles with very large magnitude motion in the videos. Second, as discussed in Sec 5.5 and Figure 6 of the paper, large occlusion events can produce ghosting artifacts. For example, in Figure 6, some cars visible in the first keyframe disappear in the second keyframe, while new cars appear in the second keyframe. In such cases, our approach attempts to blend dissimilar elements into the same region during the fusion stage, leading to artifacts in the intermediate frames where some car pixels are blended with the road pixels.
>
> > Whether the method can recover the movement such as fish turning around?
>
> We tried this, and it seems like it can not–we show the result in this [link](https://staging.dfg5r4eu5ha25.amplifyapp.com/):  the fish first swims forward, then blends with its mirrored counterpart in the middle, and then continues forward in the generated video.  Great example of a failure case, which we will include in revision.
>
> > Why not use weighted averaging?
>
> In the paper, we use the same classifier free guidance scale of 2.0 per frame, combined with a naive averaging weight of 0.5 for every frame during the fusion stage.  The effects of weighted averaging are inconclusive – linearly blending frames based on their proportional distance from each keyframe yields FVD/FID scores of 268.63/21.35 (compared to unweighted: 306.84/22.99) on Pexels, and 447.80/32.23 (compared to unweighted: 424.69/32.68) on Davis.

---

> > ### Comment · Reviewer_aMmk · 2024-11-30
> >
> > The rebuttal clarified my concerns. Considering the current method's application limitations (as mentioned in my weaknesses and questions), I maintain the score.

---

### Official Review · Reviewer_Ytxq · 2024-11-03

**Soundness:** 3
**Presentation:** 2
**Contribution:** 2
**Rating:** 6
**Confidence:** 3

**Summary:**

The paper presents Generative Inbetweening, a method for creating intermediate frames between two keyframes by adapting a pre-trained image-to-video diffusion model. This model adapts Stable Video Diffusion with dual-directional diffusion: generating video frames that interpolate both forwards and backwards in time.
This approach achieves motion-coherent inbetween frames through a technique that involves reversing the temporal self-attention maps within the U-Net model to generate backward motion from the endpoint keyframe, then combining this with forward-motion frames to produce smooth video sequences.
Evaluations on the Davis and Pexels datasets show the method’s performance against the existing techniques, including TRF and FILM, in terms of frame coherence and motion fidelity for larger motion gaps.

**Strengths:**

Strengths
The paper’s fine-tuning approach makes effective use of a pre-trained model (SVD) to generate backward motion without requiring extensive additional data or full retraining. This demonstrates an efficient approach to model adaptation.
By developing forward-backward motion consistency through temporal self-attention, the method generates smooth and coherent transitions, especially in scenarios with long differences between keyframes.
The paper provides good experimental results, using both qualitative comparisons and metrics like FID and FVD to validate performance improvements over established baselines (FILM, TRF, etc.).
Ablations explore the impact of various components and the paper transparently discusses limitations, providing clarity on the model's boundaries, especially with non-rigid motion types.

**Weaknesses:**

While the paper includes comparisons with baseline models, it lacks an in-depth discussion on the unique metrics or benchmarks used to capture differences between models, particularly in subjective aspects like motion realism. Including a more detailed discussion on why certain metrics (e.g., FID or FVD) were selected over others could clarify the relevance of the performance gains.

The model relies heavily on SVD’s motion priors, which, as the authors note, can struggle with non-rigid or complex kinematic movements.
While the paper acknowledges this, further discussion on how future models might address such limitations, possibly by incorporating other motion datasets or additional temporal constraints, would add depth to the future directions.

Although the fine-tuning approach is a strength, it may be challenging for readers unfamiliar with diffusion models to follow the model adaptation process fully. More visual aids or pseudocode detailing the fine-tuning and dual-directional sampling steps would enhance clarity.

**Questions:**

Given the model's limitations with non-rigid motions, did the authors explore any alternative solutions, such as enforcing additional temporal consistency constraints or incorporating motion priors for articulated objects?

While the quantitative results are promising, did the authors consider conducting a user study to assess perceived motion realism, as subjective assessments might capture nuances that FID/FVD cannot?

Could the authors elaborate on how sensitive the model's performance is to the choice of the 180-degree rotation in the self-attention map? Did they experiment with other configurations for reversing the temporal interaction?

---

> ### Author Response · Authors · 2024-11-26
>
> > Future work to improve non-rigid/articulated motions
>
> We agree that being able to animate articulated motion is important for generating diverse high-quality video. Lots of factors go into making this possible, but most important is the quality of the base model, so we expect this will get considerably better with subsequent models. In addition, as the reviewer mentions, including better motion datasets, and incorporating articulated motion/physical movement priors may also help. We have added this discussion to the submitted revision.
>
> > Motion realism evaluation
>
> As requested, we conducted a 28 participant user study on subsets of Davis and Pexels,where participants were shown video triplets generated with FILM, TRF, and our method with the same input frame pair, playing side-by-side in random order. Users were asked to choose which of the three videos looks most realistic. Each of the 100 examples was rated at least by one participant. Our method was chosen in 82.07% of the 714 ratings.
>
> The FVD score, which measures the distance between the distributions of generated videos and the ground-truth videos with realistic motions  should also be an indicator of motion realism, and our method outperforms the baselines on this metric too.
>
> > Rationale of 180 degree attention map rotation
>
> The 180-degree rotation flips the attention map both horizontally and vertically, causing a  reversal of the attention relationships. As noted in the ablation study (L466), solely using the rotated attention map without any fine-tuning can still generate reverse motion videos but typically suffers from poor visual quality. Other operations, such as flipping the attention map only horizontally are not able to produce videos with reverse motion.
>
> > Pseudo code
>
> We have added the Python pseudo-code for our light-weight finetuning process and dual directional diffusion sampling process in the appendix of the submitted revision to enhance the clarity of our method.

---

### Author Response · Authors · 2024-11-26

We thank the reviewers for their positive reviews and constructive avenues for improvement. We appreciate the reviewers found our work novel and efficient in adapting the image-to-video model for keyframe interpolation. We add the appendix in the submitted revision to include additional results where we add:
1. an experiment measuring our approach’s sensitivity to the scale of the training set.
2. an experiment measuring how the motion bucket ID parameter in Stable Video Diffusion affects our results
3. pseudo code for our light-weight finetuning technique and dual-directional diffusion sampling process.

We address each reviewer’s comments in the individual responses..

---

### Meta-Review · Area_Chair_yLrv · 2024-12-22

**Metareview:**

The paper presents a method to in-between the frames of a video located between two quite distant keyframes provided as an input. The reviewers acknowledged superior results compared to several recent works. At the same time, the reviewers also listed a bunch of weaknesses, such as lack of in-depth analysis of metrics to capture motion realism, lack of support for articulated motions (which the authors attributed to the use of SVD, which is known to have troubles with complex motions) and others. There was only a shallow discussion between the authors and reviewers, in which, in AC's opinion, the authors addressed only a portion of concerns. Hence, the reviewers put the paper just marginally above the borderline.

**Additional Comments On Reviewer Discussion:**

The discussion was superficial, there was no significant back and forth here, so common in other ICLR submissions. The authors addressed only a portion of concerns, the reviewers didn't provide additional questions. This is an indication to the AC the scores are appropriately set.

---

### Decision · Program_Chairs · 2025-01-22

Accept (Poster)